# Exploration of physiological and biochemical processes of canola with exogenously applied fertilizers and plant growth regulators under drought stress

Muhammad Mahran Aslam[1], Fozia Farhat[2], Mohammad Aquil Siddiqui[1]*, Shafquat Yasmeen[1], Muhammad Tahir Khan[1], Mahboob Ali Sial[1], Imtiaz Ahmad Khan[1]

1 Nuclear Institute of Agriculture (NIA), Tando Jam, Sindh, Pakistan, 2 Department of Botany, Government College Women University, Faisalabad, Pakistan

* Siddiqui_aquil@yahoo.com

**Data Availability Statement:** All relevant data are within the paper and its Supporting Information files.

## Abstract

Environmental stresses may alter the nutritional profile and economic value of crops. Chemical fertilizers and phytohormones are major sources which can enhance the canola production under stressful conditions. Physio-biochemical responses of canola altered remarkably with the use of nitrogen/phosphorus/potassium (N/P/K) fertilizers and plant growth regulators (PGRs) under drought stress. The major aim of current study was to evaluate nutritional quality and physio-biochemical modulation in canola (*Brassica napus* L.) from early growth to seed stage with NPK and PGRs in different water regimes. To monitor biochemical and physiological processes in canola, two season field experiment was conducted as spilt plot under randomized complete block design (RCBD) with four treatments (Control, Chemical fertilizers [N (90 kg/ha), P and K (45 kg ha$^{-1}$)], PGRs; indole acetic acid (IAA) 15g ha$^{-1}$, gibberellic acid (GA$_3$) 15g ha$^{-1}$ and the combination of NPK and PGRs] under different irrigations regimes (60, 100, 120, 150 mm evaporations). Water stress enhanced peroxidase (POD), catalase (CAT), superoxide dismutase (SOD), polyphenol oxidase (PPO), soluble sugar, malondialdehyde (MDA), proline contents as well as leaf temperature while substantially reduced leaf water contents (21%), stomatal conductance (50%), chlorophyll contents (10–67%), membrane stability index (24%) and grain yield (30%) of canola. However, the combined application of NPK and PGR further increased the enzymatic antioxidant pool, soluble sugars, along with recovery of leaf water contents, chlorophyll contents, stomatal conductance and membrane stability index but decreased the proline contents and leaf temperature at different rate of evaporation. There is positive interaction of applied elicitors to the water stress in canola except leaf area. The outcomes depicted that the combination of NPK with PGRs improved the various morpho-physiological as well as biochemical parameters and reduced the pressure of chemical fertilizers cost about 60%. It had also reduced the deleterious effect of water limitation on the physiology and grain yield and oil contents of canola in field experiments.

**Funding:** The author(s) received no specific funding for this work.

**Competing interests:** The authors have declared that no competing interests exist.

## 1. Introduction

Among oilseed crops, canola (*Brassica napus* L.) holds a special rank worldwide as the second highly significant oil yielding crop. with over ~40–50% oil and 40% protein (in rapeseed meal) contents after soybean [1]. Owing to continuous efforts of scientists, the production of canola had reached a milestone by enhancing its yield and area upto 68.9 million metric tons and 33.7 million hectares, respectively [2]. The leaves of this plant are imperative source of animal feed due to encompassing a balanced ratio of protein and fibrous [3]. Oil extracted from oilseed rape contains a very high concentration of unsaturated fatty acids (such as oleic acid and linoleic acid (C18:3, ~10% v/v)) making it a fatty acid-rich diet source [4]. Pakistan facing the low yield of canola somewhat due to the poor management of macro- and micronutrient. Globally, it is emerging oil seed crop among the other oil seeds due to low erucic acid and glucosinolates in oil and seed cake respectively [5].

Like many other major temperate field crops, canola is particularly susceptible to different environmental stresses, particularly heat and drought [6]. Reduced biomass and chlorophyll contents due to deterioration of chloroplast structure along with reduced production of seed oil and protein contents are usual symptoms of water stress in *B. napus* [7]. The plant growth and development mainly restricted by the drought stress via declining turgor pressure of plant cells which caused hindrance in the biochemical and physiological intriguing mechanisms [8]. The water deficit condition predominates with the reduced concentration of intracellular $CO_2$, chlorophyll destruction, photochemical system disorder and stomatal closure [9]. The plant showed biochemical response in the form of reactive oxygen species (ROS) production under drought stress. The excess amount of ROS can damage the cell membrane by elevating the lipid peroxidation [10,11]. The plants mitigate the ROS harmful effect by the activities of enzymatic and non-enzymatic antioxidants [12]. Superoxide dismutase (SOD), peroxidase (POX), catalase (CAT) and polyphenol peroxidase (PPO) are the antioxidant enzymes which diminished the ROS concentration during drought stress [10,13]. The accumulation of osmolytes, proline, soluble sugar, soluble proteins are the non-enzymatic reaction that caused osmatic regulation under water deficient condition stress [14]. The antioxidant activities elevated under stress condition in ajowan and canola plant against ROS [15]. The plants respond to drought by enhancing osmolytes (proline), antioxidant activities (POX, PPO, SOD) in plants [16,17].

N play a pivotal role in plant tissue growth and development, being an integral part of protein, chlorophyll, nucleotides, protein, amino acid which directly influence the quality and quantity of crop production [18]. The adequate N supply is important attribute to boost up the canola productivity [19]. Thus, any fluctuation in the soil profile, texture, and moisture content at various critical stage of growth and development may decrease the N use efficiency in canola. Canola crop is very responsive to fertilizer application especially N,P and K which significantly effects the growth and yield per ha [20]. It also stimulate the leaf area (LA) development after flowering in canola [21,22].

The management of soil fertility is limiting factor for the sustainable agriculture production [20]. This challenges of soil fertility can be overcome by the optimal application of fertilizers and plant growth regulators [23]. The chemical and bio-fertilizer are very effective in improving the micro- and macro nutrient via organic compound degradation and N fixation [24]. It improves nutrient uptake and reduces the damaging effects of drought in crops. This would increase the activities of PPO, CAT and POX which ultimately improve the grain yield under drought stress in response to fertilizer and PGRs [10]. The combined application of chemical fertilizers, bio fertilizers and PGRs enhanced the accumulation of proline, sugar contents and chlorophyll contents [25]. The fertilizers along with PGRs elevated the stomatal conductance,

water contents and total chlorophyll content under water limiting conditions [26]. These combined application of nutrients improve the soil fertility under water stress condition [27]. Although genetic manipulation has a promising effects to mitigate this problem and also stated mid- and long-term results, but the current demands of food and feed required some immediate response methods to address food security and hidden hunger. In spite of numerous scientific efforts, information related to biochemical, physiological and yield dynamics with respect to combined application of chemical fertilizers and PGRs is very limited for canola production. Thus, this manuscript comprised to gauge the morphological, physiological and biochemical rejoinders of canola to assimilate fertilizers management and PGRs with limited supply of water.

## 2. Materials and methods

### 2.1. Experimental design and treatments

The two-year (2018–19 & 2019–2020) field experiment were conducted at experimental field of Nuclear Institute of Agriculture (NIA), TandoJam, Sindh to investigate the variability in physio-biochemical parameters and yield of canola (Surhan-2012) in response to fertilization under excessive rate of water evaporation. The source of N, P and K in the current experiment was ammonium nitrate ($NH_4NO_2$), single super phosphate (SSP) and potassium sulphate ($K_2SO_4$), respectively. NPK fertilizers were applied in the rhizosphere of canola growing field at the time of sowing and PGRs were applied at its flowering stage. IAA and $GA_3$ was thoroughly dissolved in dimethyl sulfoxide (DMSO) along with tween-20 (T-20) as surfactant. The experiment was performed in split plots as random complete block design (RCBD) in triplicate with four irrigation levels. Weather data was carefully monitored from October to April for both growing years (2018-19/2019-20). The maximum and minimum temperature decreases from October to March and then increases marginally in April for both growing seasons (Table 1). The soil of experimental area was analyzed and found 2% organic matter (OM) along with sodium (0.08), potassium (0.11), magnesium (3.2), sulpher (16.90), zinc (1.40), boron (0.34), phosphorus (10.83) and calcium (2.5) as major nutrients. The soil mineral contents were calculated as µg/g soil (Table 2). The detail of treatment is given in the table below.

Weeding operations was done frequently during growth and development of surhan-2012 plants in both years.

### 2.2. Determination of agronomic and yield related parameters

At vegetative stage, height of ten randomly selected canola plants of each treatment were measured with measuring tape (cm) from ground to the tip of flag leaf and counted number of branches per plant while number of siliqua plant$^{-1}$, siliqua length (cm) and number of seeds

**Table 1. Weather data of experimental location during 2016–17.**

| Month | $T_{max}$ $^0$C | | $T_{min}$ $^0$C | | Total Rain fall (mm) | | Relative Humidity (%) | |
|---|---|---|---|---|---|---|---|---|
| | 2018–19 | 2019–20 | 2018–19 | 2019–20 | 2018–19 | 2019–20 | 2018–19 | 2019–20 |
| October | 37.1 | 38.4 | 20.6 | 19.5 | 0 | 0 | 57 | 53 |
| November | 32.6 | 31.20 | 12.9 | 12.6 | 0 | 0 | 52 | 51 |
| December | 28.5 | 25.6 | 11.2 | 8.0 | 0 | 0 | 56 | 54 |
| January | 25.4 | 22.1 | 10.3 | 7.3 | 0 | 2.0 | 60 | 61 |
| February | 21.4 | 28.2 | 9.2 | 10.0 | 0 | 0 | 44 | 49 |
| March | 34.1 | 34.4 | 17.1 | 14.5 | 0 | 0 | 47 | 47 |
| April | 39.0 | 40.4 | 20.9 | 19.9 | 0 | 0 | 43 | 42 |

**Table 2. Soil properties of experimental field.**

| Site | OM (%) | pH | meq/100g soil µg/g soil | | | | | | | |
|---|---|---|---|---|---|---|---|---|---|---|
| **NIA, Tandojam Exp. Farm** | 2 | 6.5–7.0 | Na | K | Mg | S | Zn | B | P | Ca |
| | | | 0.08 | 0.11 | 3.2 | 16.90 | 1.40 | 0.34 | 10.83 | 2.5 |

| Irrigation Levels | Treatment with Elicitors |
|---|---|
| $l_0$ = 60mm evaporation | $T_0$ = No treatment<br>$T_1$ = Chemical Fertilizers [N (90 kg ha$^{-1}$), P (45 kg ha$^{-1}$) and K (45 kg ha$^{-1}$)]<br>$T_2$ = PGRs [IAA (15g ha$^{-1}$) and GA$_3$ (15g ha$^{-1}$)]<br>$T_3$ = $T_1$ + $T_2$ |
| $l_1$ = 100mm evaporation | $T_0$ = No treatment<br>$T_1$ = Chemical Fertilizers [N (90 kg ha$^{-1}$), P (45 kg ha$^{-1}$) and K (45 kg ha$^{-1}$)]<br>$T_2$ = PGRs [IAA (15g ha$^{-1}$) and GA$_3$ (15g ha$^{-1}$)]<br>$T_3$ = $T_1$ + $T_2$ |
| $l_2$ = 120mm evaporation | $T_0$ = No treatment<br>$T_1$ = Chemical Fertilizers [N (90 kg ha$^{-1}$), P (45 kg ha$^{-1}$) and K (45 kg ha$^{-1}$)]<br>$T_2$ = PGRs [IAA (15g ha$^{-1}$) and GA$_3$ (15g ha$^{-1}$)]<br>$T_3$ = $T_1$ + $T_2$ |
| $l_3$ = 150mm evaporation | $T_0$ = No treatment<br>$T_1$ = Chemical Fertilizers [N (90 kg ha$^{-1}$), P (45 kg ha$^{-1}$) and K (45 kg ha$^{-1}$)]<br>$T_2$ = PGRs [IAA (15g ha$^{-1}$) and GA$_3$ (15g ha$^{-1}$)]<br>$T_3$ = $T_1$ + $T_2$ |

siliqua$^{-1}$ were counted after harvesting. Similarly, recorded the data related to days to flower when 50% flowers had been appeared on plants and days to maturity at discoloration stage. Grains of canola plants were dried in the sun and recorded 1000 grain weight by using digital balance (Model- Explorer OHAUS). Biological yield was determined after harvesting from 4 rows and calculated as kg ha$^{-1}$.

## 2.3. Determination of quality traits

Oil content were extracted with petroleum ether using soxhlet apparatus. All the dried seed samples were coarsely ground and packed carefully into the thimble for oil extraction. The extraction was performed continuously for three cycles (90–120 min.) and oil productivity was drawn through standard formula. Two significant fatty acids, erucic acid (%) and glusinolate (µmol g$^{-1}$) were detected through High Performance Liquid Chromatography (HLPC). Oil contents were analysed using Gradient HPLC (Shimadzu, Japan) having LC-10AT, SCTL 10A system controller, SPD-10AR UV-VIS detector at 280 nm with C18 stationary column (Shim-Pack CLC-ODS). Elution was done for 60 min with a flow rate of 1ml/min in a gradient system of two mobile phases A ($H_2O_2$: AA-94:6, pH 2.27), B (ACN100%) [28]. Moisture contents (%) was determined by the weight of water in a seed.

## 2.4. Determination of NPK uptake by the canola grain

For determination of P and K contents, seed samples of each treatment were dried at 70°C for 48h. Dried and powdered grain sample (0.5g) was digested with 20mL concentrated nitric acid ($HNO_3$) by adopting method of Rathje and Jackson [29]. The samples were placed for 3 hours at room temperature. After 3 hours, samples were laid on the digestion block at 250°C until the solution became tinted yellow in appearance. The digested solution was diluted with 50mL of distilled water and filtered with whatman No. 42 filter paper. The P contents from the digested plant samples was determined by recording optical density at 430 nm with spectro-photometer (Model-Spectronic-21) by Primson et al. [30]. The K content in gains was resolute

by flame photometer (Model-Flame photometer-400) according to the method suggested by Tammam [31]. Nitrogen content was examined by Kjeldahl apparatus [32]. Following formulas were applied to determined NPK uptake in grains of canola as kg ha$^{-1}$.

N uptake (kg ha$^{-1}$) in grain: N (%) x grain yield (kg ha$^{-1}$) /100
P uptake (kg ha$^{-1}$) in grain: P (%) x grain yield (kg ha$^{-1}$) /100
K uptake (kg ha$^{-1}$) in grain: K (%) x grain yield (kg ha$^{-1}$) /100

## 2.5. Determination of physiological and biochemical parameters

**2.5.1. Chlorophyll contents.** Fresh leaf samples were collected from each treatment and subjected to grinding with 80% acetone. Semi-liquid extract was filtered and centrifuged at 10000rpm for 5minutes [33]. The supernatant was then subjected to spectrophotometer (Model Analytikjena Spekol 1500 Germany)

**2.5.2. Leaf water content.** Leaf water content was measured by harvesting three leaves per plant from every plot after 45 days of sowing (DAS). Fresh leaf sample was weighed in gram (g) as fresh weight (FW) and let them dry at high temperature (80°C) and reweighed as dry weight (DW). Leaf water content (LWC) was calculated by following formula.

$$LWC = [(FW - DW)/(FW)]$$

**2.5.3. Leaf temperature.** The leaf temperature (LT) was measured at flowering stage with the help of infrared thermometer (TES- 1327). The leave temperature ($^0$C) was measured by randomly selecting 3 plants of every treatment and replicate. Later, the mean LT was carefully recorded.

**2.5.4. Stomatal conductance (g$_s$).** Portable photosystem (Porometer AP4, Delta-T Devices Ltd., Cambridge, U.K.) was used to measure the stomatal conductance. This data was carefully recorded 60 days after sowing (DAS). This measurement was carried out from 10:00 to 14.00 h.

**2.5.5. Membrane stability index (MSI).** The previously reported method of Ghassemi-Golezani et al. [34] with slight modification was used to calculate membrane stability index. ˜0.1 g leaf samples was mixed with double distilled water (10 ml) in falcon tube and incubated at 40°C for 30 min and electrical conductivity was measured (EC$_1$). Thereafter conductivity of these sample were assessed after placing water bath at 100°C for 10 min (EC$_2$). The MSI was measured by the following formulas:

$$MSI = (EC_1/EC_2) \times 100$$

## 2.6. Determination of osmolytes

The total soluble sugar content was estimated from the dried leaves of all the replicates of respective canola treatments [35]. The standard calibration curve of pure glucose was used to determine total soluble sugars of leaves and expressed as mg/g DW. To determine proline contents in canola, leaf sample was thoroughly grinded in 3% sulfosalicylic acid. The extracted sample was filitered, mixed with glacial acetic acid and ninhydrin in a test tube with a ratio of 1:1:1. This mixture was heated at 100°C for 60 min in a Bain Marie oven. Then reaction mixture was cooled at room temperature and the toluene used for the extraction of mixture, vortexed for 30sec. The absorbance of the upper organic phase was recorded at 520 nm. Calibration curve of pure proline was used to compare the proline content of canola leaves and expressed as mg/g FW [36].

## 2.7. Determination of antioxidants

Young leaves were collected from each treatment at 60 DAS and assayed the activity of polyphenol oxidase (PPO) by Kumar and Khan (1982) method [37]. The reaction mixture contains 0.1 M phosphate buffer (pH 7.8), 1 ml catechol and 5 ml enzyme extract. The reaction mixture was incubated at 25° C for 5 min, later, the reaction was terminated by dissolving 1 ml of 3 ml $NH_2SO_4$. The PPO activity was determined in the form of absorbance of resultant purpurogallin at 495 nm and expressed as $Umg^{-1}$ (U = change in 0.1 absorbance $min^{-1}$, $mg^{-1}$ protein. The CAT activity was determined with an interval of 20 seconds for 2 minutes at 240 nm ($Ug^{-1}$ FW) according to the devised method of Singh and Sharma [38]. The POX activity was observed with an interval of 30 sec for 2 minutes at 470 nm due to guaiacol oxidation. The activity was determined from reaction solution consisted 1 ml of 1% guaiacol, 0.3 ml of enzyme extract, 2.5 ml of 50 mM potassium buffer (pH = 7.0) and 1 ml of 1% $H_2O_2$ for 2 min in reaction mix [39]. The SOD activity was assessed by the estimation of volume of enzyme affected as 50% inhibition of nitroblue tetrazolium [39].

## 2.8. Determination of lipid peroxidation

Malondialdehyde content (mmol $g^{-1}$ FW) from canola leaves was determined 60 DAS to estimate rate of lipid peroxidation [40]. ˜0.5 g of fresh leaves was homogenized in 5% trichloroacetic acid (5 ml), heated at 25° C for 10 minutes and centrifuged at 1800g. The 2-thiobarbituric acid (TBA) was added in supernatant, placed at 98° C for 10 min and cooled at room temperature. Finally, recorded the absorbance at 532 nm with spectrophotometer.

## 2.9. Statistical analysis

All the experimental data was recorded and subjected to analysis of variance (ANOVA) with linear models of statistics to observe statistical significant/non-significant differences among different traits of *Brassica napus* through computer program, Student Edition of Statistix (SWX), Version 8.1 (Analytical Software, 2005). Moreover, least significant difference (LSD) test was applied to verify the level of significance (5%) among different combination means [41].

## 3. Results

The results of canola presented in this manuscript was recorded for two consecutive years i.e. 2018–2019 and 2019–2020. The mean of all attributes have been tabulated and described in the result section (Tables 3–7).

## 3.1. Agronomic and yield performance of canola with fertilizers and PGRs under drought

The mean data of two consecutive years of agronomic as well as yield attributes (plant height, days to flower, number of branches per plant, number of seed per plant, biological yield per plot, 1000 seed weight and seed yield) presented a significant (p<0.01) interaction of irrigation to that of NPK and PGR (Table 3). The plant growth was affected by the severe water stress ($I_4$), when no elicitor was provided to the canola seedlings. Plant height, leaf area and number of branches per plant decreased upto 2, 5 and 15% at maximum level of evaporation. The combination of NPK and PGRs ($T_3$) enhanced the agronomic performance under severe water deficit ($I_3$ = 150mm evaporation) condition by improving number of seeds/plant (1.76), biological yield/plant (5.32kg), 1000 seed weight (4.32g) and seed yield/hectare (2318kg/ha) (Table 3). It was observed that days to flower decreased upto 1% and 8.82% with NPK and

**Table 3. Variation in agronomical and yield attributes of canola with fertilizers and plant growth regulators under different water regimes.**

| Irrigation | Treatment with Elicitors | Plant Height (cm) | Number of branches/plant | Leaf Area ($cm^2$) | Days to Flower | Number of seeds/plant | Biological yield/ Plant (Kg) | 1000 seed weight(g) | Seed yield (kg ha$^{-1}$) | Grain Yield (kg ha$^{-1}$) |
|---|---|---|---|---|---|---|---|---|---|---|
| $I_0$ Normal | $T_0$ | 114$^c$ | 5.60$^d$ | 90.5$^d$ | 68$^b$ | 160$^d$ | 5.30$^d$ | 3.98$^d$ | 1804$^d$ | 188$^g$ |
| | $T_1$ | 114.5$^b$ | 5.73$^c$ | 91.3$^c$ | 68$^b$ | 171$^c$ | 5.38$^c$ | 4.28$^c$ | 2073$^b$ | 264$^a$ |
| | $T_2$ | 115$^a$ | 5.78$^b$ | 92.4$^b$ | 69$^a$ | 176$^b$ | 5.36$^b$ | 4.33$^b$ | 2060$^c$ | 220$^d$ |
| | $T_3$ | 113$^d$ | 5.85$^a$ | 94.0$^a$ | 66$^c$ | 188$^a$ | 5.43$^a$ | 4.96$^a$ | 2450$^a$ | 249$^b$ |
| $I_1$ Mild evaporation | $T_0$ | 113$^b$ | 5.66$^d$ | 89.0$^d$ | 70$^a$ | 164$^d$ | 5.6$^a$ | 4.0$^d$ | 1800$^d$ | 176.4$^h$ |
| | $T_1$ | 114$^a$ | 5.81$^b$ | 90.0$^c$ | 67$^c$ | 170$^b$ | 5.20$^b$ | 4.15$^c$ | 2096$^c$ | 232.42$^c$ |
| | $T_2$ | 114$^a$ | 5.80$^c$ | 90.5$^b$ | 68$^b$ | 169$^c$ | 5.16$^d$ | 4.19$^b$ | 2118$^b$ | 207.24$^e$ |
| | $T_3$ | 113$^b$ | 5.83$^a$ | 91.0$^a$ | 65$^d$ | 178$^a$ | 5.29$^c$ | 4.23$^a$ | 2340$^a$ | 221.18$^{bc}$ |
| $I_2$ Moderate evaporation | $T_0$ | 112$^b$ | 5.58$^d$ | 86.4$^d$ | 70$^a$ | 165$^d$ | 4.94$^d$ | 3.90$^d$ | 1650$^e$ | 131.06$^k$ |
| | $T_1$ | 112$^b$ | 5.78$^c$ | 89.5$^c$ | 65$^c$ | 169$^c$ | 5.14$^c$ | 4.18$^b$ | 1943$^b$ | 165.97$^i$ |
| | $T_2$ | 113$^a$ | 5.80$^b$ | 90.0$^b$ | 67$^b$ | 172$^b$ | 5.19$^b$ | 4.09$^c$ | 1940$^c$ | 172.03$^h$ |
| | $T_3$ | 112$^b$ | 5.82$^a$ | 93.0$^a$ | 64$^d$ | 178$^a$ | 5.27$^a$ | 4.26$^a$ | 2353$^a$ | 198.21$^f$ |
| $I_3$ Severe evaporation | $T_0$ | 113$^a$ | 5.52$^d$ | 85.8$^d$ | 69$^a$ | 150$^d$ | 3.89$^d$ | 3.67$^d$ | 1538$^f$ | 170.52$^h$ |
| | $T_1$ | 112$^b$ | 5.68$^c$ | 88.4$^b$ | 67$^c$ | 168$^b$ | 5.16$^b$ | 4.13$^c$ | 1923$^c$ | 173.23$^h$ |
| | $T_2$ | 113$^a$ | 5.70$^b$ | 87.0$^c$ | 68$^b$ | 163$^c$ | 5.14$^c$ | 4.15$^b$ | 1975$^b$ | 133.05$^k$ |
| | $T_3$ | 111$^c$ | 5.79$^a$ | 92.3$^a$ | 62$^d$ | 176$^a$ | 5.32$^a$ | 4.26$^a$ | 2318$^a$ | 156.84$^j$ |
| $F_{test}$ | I × T | 1.23* | 2.65** | 0.023ns | 7.75* | 68.16** | 1.27** | 0.53* | 30.64** | 1803.26** |

Note: $I_0$ = 60mm evaporation, $I_1$ = 100mm evaporation, $I_2$ = 120mm evaporation, $I_3$ = 150mm evaporation, $T_0$ = No treatment

$T_1$ = NPK, $T_2$ = PGRs, $T_3$ = ($T_1$+$T_2$). The alphabetical superscript in a column present significant difference among the treatments to different rate of evaporation from highest to lowest value (a = highest value).

* = least significant

**significant

*** highly significant.

PGRs respectively under severe rate of evaporation ($I_3$). However, number of seeds per plant, biological yield and seed yield increased upto 11%, 0.4% and 28% respectively with $T_3$ treatment (NPK and PGRs) at maximum rate of evaporation. Grain yield was recovered with NPK (23%), PGRs (10%) and their combined treatment (17%) under least rate of evaporation ($I_1$) compared to reduction caused in non-treated canola plants ($T_0$/6%). The low water supply during critical growth stage reduced the yield of canola (Table 3). Moreover, PGRs showed non-significant difference among all rates of evaporation for plant height but significantly vary for other agronomic and yield traits. The fertilizer applications significant enhanced the biological seed yield/plant (5.43kg) of canola and the highest seed yield (2450kg/ha) was recorded with NPK ($T_1$) under normal rate of evaporation ($I_0$). The $T_1$ and $T_2$ treatment presented a non- significant difference under normal irrigation condition. Moreover, data displayed a strong and significant interaction between different rate of evaporation (I) and applied elicitors (T) for all studied morphological and yield related features of canola except leaf area (Table 3).

## 3.2. Physiological performance of canola with fertilizers and PGRs under drought

The mean data of two-year field experiment of canola revealed a highly significant (p<0.01) response of NPK and PGRs application to chlorophyll contents under water deficit condition (Table 4). The chlorophyll contents decline (10–67%) significantly in canola with increasing rate of evaporation ($I_0$-$I_3$). However, the exogenous application of NPK ($T_1$) significantly

**Table 4. Variation in physiological attributes of canola with chemical fertilizers and plant growth regulators under different water regimes.**

| Irrigation | Treatments | Chl a | Chl b | Total Chlorophyll | LWC | LT | Stomatal Conductance |
|---|---|---|---|---|---|---|---|
| | | (mg/g FW) | | | (%) | ($^0$C) | (mmol m$^{-2}$ s$^{-1}$) |
| $I_0$ Normal | $T_0$ | 1.43$^f$ | 0.834$^c$ | 2.264$^d$ | 80.0$^{ab}$ | 21.8$^i$ | 142.3$^c$ |
| | $T_1$ | 2.18$^a$ | 0.87$^a$ | 3.05$^a$ | 83.0$^a$ | 19.7$^l$ | 146.2$^a$ |
| | $T_2$ | 1.63$^e$ | 0.84$^b$ | 2.47$^d$ | 82.0$^a$ | 21.6$^j$ | 144. 1$^b$ |
| | $T_3$ | 1.95$^b$ | 0.85$^b$ | 2.8$^c$ | 81.9$^a$ | 19.8$^k$ | 146.42$^a$ |
| $I_1$ Mild evaporation | $T_0$ | 1.28$^h$ | 0.78$^d$ | 2.06$^f$ | 80.82$^{ab}$ | 25.6$^f$ | 140.73$^d$ |
| | $T_1$ | 1.88$^c$ | 0.82$^c$ | 2.7$^c$ | 81.80$^a$ | 22.3$^h$ | 145.12$^a$ |
| | $T_2$ | 1.48$^f$ | 0.80$^c$ | 2.28$^d$ | 81.17$^a$ | 23.5$^g$ | 144.95$^b$ |
| | $T_3$ | 1.72$^d$ | 0.80$^c$ | 2.52$^b$ | 82.0$^a$ | 21.8$^i$ | 143.91$^b$ |
| $I_2$ Moderate evaporation | $T_0$ | 0.73$^k$ | 0.72$^f$ | 1.45$^i$ | 74.17$^{de}$ | 30.6$^c$ | 101.0$^i$ |
| | $T_1$ | 1.18$^i$ | 0.75$^e$ | 1.93$^g$ | 76.16$^{cd}$ | 28.5$^d$ | 107.9$^h$ |
| | $T_2$ | 1.18$^i$ | 0.75$^e$ | 1.93$^g$ | 78.18$^{bc}$ | 27.6$^e$ | 117.26$^f$ |
| | $T_3$ | 1.39$^g$ | 0.76$^e$ | 2.15$^e$ | 78.84$^{bc}$ | 25.91$^f$ | 119.47$^e$ |
| $I_3$ Severe evaporation | $T_0$ | 0.47$^m$ | 0.68$^g$ | 1.15$^l$ | 62.49$^f$ | 35.0$^a$ | 70.94$^k$ |
| | $T_1$ | 0.66$^l$ | 0.71$^f$ | 1.37$^k$ | 65.18$^f$ | 33.4$^b$ | 78.61$^j$ |
| | $T_2$ | 0.85$^k$ | 0.70$^f$ | 1.55$^j$ | 71.80$^e$ | 28.92$^d$ | 97.67$^i$ |
| | $T_3$ | 1.13$^j$ | 0.72$^f$ | 1.85$^h$ | 73.6$^{de}$ | 27.9$^e$ | 110.4$^g$ |
| F $_{test}$ | I × T | 0.134** | 0.00032** | 1.982** | 0.139** | 21.99** | 143.4** |

Note: $I_0$ = 60mm evaporation, $I_1$ = 100mm evaporation, $I_2$ = 120mm evaporation, $I_3$ = 150mm evaporation, $T_0$ = No treatment

$T_1$ = NPK, $T_2$ = PGRs, $T_3$ = ($T_1$+$T_2$). The alphabetical superscript in a column present significant difference among the treatments to different rate of evaporation from highest to lowest value (a = highest value).

* = least significant

**significant

*** highly significant.

enhanced *Chl a*, *Chl b* and total chlorophyll contents under control but combination of NPK and PGRs ($T_3$) progressively recovered the chlorophyll contents from least to severe water stress ($I_1$-$I_3$). The *Chl a* contents decreased significantly with progression of water stress, even fortification of NPK and PGRs failed to completely mitigate adverse effects of severe rate of evaporation. A highly significant relation was observed between different rate of evaporation (I) and treatments with elicitors (T) for chlorophyll contents and also showed a recovery mechanism by promoting photosynthetic activity (Table 4).

The average of two season's data for water contents and temperature of canola leaves showed a significant (p<0.01) interaction of irrigation regimes and elicitors (Table 4). Both these traits worked antagonistically as decrease in leaf water content (LWC) ensured increased leaf temperature (LT) under severe drought ($I_3$) effect. Leaf temperature enhanced (40-6-%) in parallel to decrease in LWC (7–21%) with moderate ($I_2$) and severe rate of evaporation ($I_3$) but NPK and PGRs minimized the impact of evaporation and enhanced these features compared to their stress condition (Table 4). The irrigation regimes $I_0$ and $I_1$ showed non-significant difference on water contents and leaf temperature and same effects were observed with NPK ($T_1$) under mild ($I_1$) and moderate ($I_2$) rate of evaporation (Table 4).

The NPK application significantly (p<0.01) influenced the stomatal conductance of canola plant with different rate of evaporation (Table 4). The water stress reduced the stomatal conductance (50%) under severe rate of evaporation ($I_4$) but exhibited non-significant difference under the mild ($I_1$) and moderate ($I_2$) rate of evaporation. The combined effect of PGRs and NPK ($T_3$) enhanced (1–2%) the stomatal conductance under normal ($I_0$) and mild ($I_1$) water

**Table 5. Variation in quality traits of canola with chemical fertilizers and plant growth regulators under different water regimes.**

| Irrigation | Treatments | Oil content (%) | Oil yield (kg ha$^{1}$) | Protein (%) | Moisture (%) | Glucosionalate (μmol/g) | Erucic acid (%) |
|---|---|---|---|---|---|---|---|
| $I_0$ Normal | $T_0$ | 38.0$^d$ | 238.0$^d$ | 20.77$^d$ | 5.23$^d$ | 26.2$^a$ | 4.44$^a$ |
| | $T_1$ | 38.5$^c$ | 324.5$^c$ | 24.82$^c$ | 5.72$^c$ | 22.0$^c$ | 4.20$^b$ |
| | $T_2$ | 39.0$^b$ | 328.9$^b$ | 24.97$^b$ | 5.96$^b$ | 18.7$^d$ | 3.74$^c$ |
| | $T_3$ | 42.8$^a$ | 468.0$^a$ | 25.07$^a$ | 6.77$^a$ | 14.6$^f$ | 3.22$^d$ |
| $I_1$ Mild evaporation | $T_0$ | 37.0$^d$ | 234.6$^d$ | 20.0$^d$ | 5.0$^d$ | 24.0$^b$ | 5.21$^a$ |
| | $T_1$ | 37.8$^c$ | 340.0$^c$ | 24.82$^c$ | 5.36$^b$ | 20.0$^c$ | 4.37$^b$ |
| | $T_2$ | 38.2$^b$ | 355.0$^b$ | 24.79$^b$ | 5.30$^c$ | 19.0$^d$ | 3.98$^c$ |
| | $T_3$ | 42.5$^a$ | 437.8$^a$ | 25.0$^a$ | 5.98$^a$ | 16.0$^e$ | 3.41$^d$ |
| $I_2$ Moderate evaporation | $T_0$ | 36.0$^d$ | 229.0$^d$ | 18.5$^d$ | 4.93$^c$ | 23.5$^b$ | 4.98$^a$ |
| | $T_1$ | 37.3$^c$ | 324.5$^c$ | 23.0$^c$ | 5.11$^c$ | 17.0$^e$ | 3.75$^c$ |
| | $T_2$ | 37.9$^b$ | 329.38$^b$ | 23.5$^b$ | 5.19$^b$ | 18.2$^d$ | 3.82$^b$ |
| | $T_3$ | 41.6$^a$ | 398.0$^a$ | 24.8$^a$ | 5.75$^a$ | 14.0$^f$ | 3.45$^d$ |
| $I_3$ Severe evaporation | $T_0$ | 34.8$^d$ | 220.0$^d$ | 18.0$^d$ | 4.81$^d$ | 20.0$^c$ | 4.99$^a$ |
| | $T_1$ | 36.0$^c$ | 378.3$^c$ | 23.0$^b$ | 5.25$^c$ | 18.0$^d$ | 3.68$^c$ |
| | $T_2$ | 36.5$^b$ | 382.0$^b$ | 22.5$^c$ | 5.29$^b$ | 16.5$^e$ | 3.71$^b$ |
| | $T_3$ | 41.0$^a$ | 437.2$^a$ | 24.2$^a$ | 5.78$^a$ | 11.6$^g$ | 3.50$^d$ |
| $F_{test}$ | I × T | 13.68** | 6.34* | 3.36** | 0.985** | 4.28* | 12.78** |

Note: $I_0$ = 60mm evaporation, $I_1$ = 100mm evaporation, $I_2$ = 120mm evaporation, $I_3$ = 150mm evaporation, $T_0$ = No treatment

$T_1$ = NPK, $T_2$ = PGRs, $T_3$ = ($T_1+T_2$). The alphabetical superscript in a column present significant difference among the treatments to different rate of evaporation from highest to lowest value (a = highest value).

* = least significant

**significant, *** highly significant.

stress. A drastic reduction of stomatal conductance was recorded with increasing water stress. A significant interaction of stomatal conductance (143.4**) was recorded for rate of evaporation and applied treatments (Table 4).

### 3.3. Qualitative performance of canola with fertilizers and PGRs under drought

The mean of two years statistical data depicted that the interactive effect of irrigation regimes and fertilizers significantly influenced the quality related traits (oil contents, oil yield, protein, glucosinolates, and moisture and erucic acid contents) of canola (Table 5). These features declined with the progression of water deficit condition. The experimental results defined the positive influence of NPK and PGRs ($T_3$) on oil yield (67–83%), oil (7–11%), protein (16–20%) and moisture (9–14%) contents while reduced glucosinolates (38–55%) and erucic acid contents (20–22%) with ongoing increasing rate of evaporation ($I_1$-$I_3$) (Table 5). It is summarized that $T_3$ treatment is a good rehabilitation strategy to improve quality of canola followed by $T_1$ and $T_2$ to address water scarcity issues of canola.

### 3.4. Nitrogen, phosphorus and potassium (NPK) contents and their uptake in canola

The mean data of two consecutive years (2018-19/2019-20) showed that combined effect of fertilizer and PGRs ($T_3$) significantly (p<0.01) influence the NPK contents and their uptake in canola plant. Provision of NPK ($T_1$) individually or in combination with PGRs ($T_3$) improved the NPK contents as well as their uptake under mild ($I_1$) irrigation stress (Table 6). The N

**Table 6. Variation in nitrogen (n), phosphorus (p) and potassium (k) of canola with chemical fertilizers and plant growth regulators under different water regimes.**

| Irrigation | Treatments | N (%) | P (%) | K (%) | N uptake (kg ha$^1$) | P uptake (kg ha$^1$) | K uptake (kg ha$^1$) |
|---|---|---|---|---|---|---|---|
| I$_0$ Normal | T$_0$ | 2.70$^d$ | 0.18$^d$ | 1.21$^d$ | 22.81$^d$ | 17.43$^d$ | 16.34$^d$ |
| | T$_1$ | 3.72$^b$ | 0.21$^b$ | 1.68$^b$ | 38.45$^b$ | 21.09$^b$ | 18.23$^b$ |
| | T$_2$ | 3.59$^c$ | 0.20$^c$ | 1.55$^c$ | 34.23$^c$ | 19.94$^c$ | 17.78$^c$ |
| | T$_3$ | 3.88$^a$ | 0.24$^a$ | 1.88$^a$ | 51.84$^a$ | 23.29$^a$ | 22.67$^a$ |
| I$_1$ Mild evaporation | T$_0$ | 2.69$^d$ | 0.20$^c$ | 1.20$^d$ | 22.86$^d$ | 18.43$^d$ | 16.0$^d$ |
| | T$_1$ | 3.35$^c$ | 0.20$^c$ | 1.55$^b$ | 38.98$^b$ | 20.45$^b$ | 18.45$^b$ |
| | T$_2$ | 3.54$^b$ | 0.21$^b$ | 1.43$^c$ | 29.74$^c$ | 19.90$^c$ | 18.21$^c$ |
| | T$_3$ | 3.75$^a$ | 0.22$^a$ | 1.78$^a$ | 50.95$^a$ | 22.45$^a$ | 21.45$^a$ |
| I$_2$ Moderate evaporation | T$_0$ | 2.71$^d$ | 0.19$^c$ | 1.16$^d$ | 20.09$^d$ | 17.09$^d$ | 14.76$^d$ |
| | T$_1$ | 3.15$^c$ | 0.22$^a$ | 1.61$^b$ | 30.56$^b$ | 19.90$^b$ | 17.85$^c$ |
| | T$_2$ | 3.19$^b$ | 0.21$^b$ | 1.34$^c$ | 26.64$^c$ | 18.64$^c$ | 17.93$^b$ |
| | T$_3$ | 3.84$^a$ | 0.21$^b$ | 1.72$^a$ | 49.80$^a$ | 21.98$^a$ | 20.06$^a$ |
| I$_3$ Severe evaporation | T$_0$ | 2.55$^d$ | 0.17$^c$ | 1.03$^d$ | 17.65$^d$ | 15.43$^d$ | 13.49$^d$ |
| | T$_1$ | 3.38$^c$ | 0.21$^b$ | 1.45$^c$ | 30.32$^c$ | 18.45$^c$ | 17.83$^c$ |
| | T$_2$ | 3.45$^b$ | 0.21$^b$ | 1.54$^b$ | 33.90$^b$ | 19.05$^b$ | 16.98$^c$ |
| | T$_3$ | 3.78$^a$ | 0.24$^a$ | 1.69$^a$ | 48.73$^a$ | 20.97$^a$ | 19.56$^a$ |
| F $_{test}$ | I × T | 3.05** | 0.168*** | 0.98** | 16.24* | 9.34*** | 3.44** |

Note: I$_0$ = 60mm evaporation, I$_1$ = 100mm evaporation, I$_2$ = 120mm evaporation, I$_3$ = 150mm evaporation, T$_0$ = No treatment, T$_1$ = NPK, T$_2$ = PGRs, T$_3$ = (T$_1$+T$_2$). The alphabetical superscript in a column present significant difference among the treatments to different rate of evaporation from highest to lowest value (a = highest value).

* = least significant

**significant

*** highly significant.

uptake and percentage (%) accumulation has been decreased upto 22 and 5% while P uptake and % accumulation decreased upto 11 and 3% respectively, and K uptake and % accumulation halso reduced upto 17 and 14% respectively, at severe rate of evaporation (I$_3$). But plants treated with NPK and PGRs (T$_3$) enhanced overall nutrient pool (N, P, K) at all rate of evaporation (I$_1$-I$_3$)

## 3.5. Biochemical performance of canola with fertilizers and PGRs under drought

The results obtained from two-year trails illustrated that osmolytes considerably (p<0.01) influenced by the applied elicitors under different irrigation regimes (I$_0$-I$_3$). The concentration of proline and total soluble sugar elevated upto 6–12% and 4–30 in response to mild to severe rate of evaporation in canola plant as an innate response mechanism. Further, the application of NPK and PGRs (T$_3$) showed a negative impact on the proline accumulation but positively enhanced soluble sugar contents with progression of evaporation rate (Table 7). The application of NPK and PGRs (T$_3$) showed 115% increase for TSS contents at severe rate of evaporation (I$_3$). A similar trend for TSS was observed with mild to moderate rate of evaporation by the application of NPK and PGRs (T$_3$).

The antioxidant enzymes and MDA activities significantly (p<0.01) influenced by the irrigation (I) and elicitors (T) in canola (Table 7). The canola plant enhanced the activities of various antioxidants and enzymes including SOD (138–643%), CAT (182–1147%), PPO (67–304%), POX (134–1752%) and MDA (47–208%) contents in response to limited water supply (I$_1$-I$_3$). The canola behavior with fertilizers did not show significant effect during normal (I$_0$)

**Table 7. Variation in biochemical attributes of canola with chemical fertilizers and plant growth regulators under different water regimes.**

| Irrigation | Treatments | PPO | CAT | POX | SOD | MDA | Soluble Sugar | Proline | MSI |
|---|---|---|---|---|---|---|---|---|---|
| | | (U g$^{-1}$ FW) | | | | (mmol g$^{-1}$ FW) | (mg /g DW) | (mmol g$^{-1}$ FW) | |
| I$_0$ Normal | T$_0$ | 0.47$^f$ | 0.21$^h$ | 0.15$^f$ | 0.26$^g$ | 2.4$^j$ | 31.4$^{ijk}$ | 15.4$^{ij}$ | 86.71$^b$ |
| | T$_1$ | 0.54$^f$ | 0.23$^h$ | 0.17$^f$ | 0.31$^{fg}$ | 2.29$^j$ | 30.0$^k$ | 16$^{ij}$ | 87.92$^a$ |
| | T$_2$ | 0.52$^f$ | 0.26$^h$ | 0.16$^f$ | 0.33$^{fg}$ | 2.28$^j$ | 31.0$^{ijk}$ | 15.03$^{ij}$ | 87.81$^a$ |
| | T$_3$ | 0.57$^f$ | 0.24$^h$ | 0.18$^f$ | 0.32$^{fg}$ | 2.27$^j$ | 30.2$^{jk}$ | 15.07$^{ij}$ | 87.45$^a$ |
| I$_1$ Mild evaporation | T$_0$ | 0.79$^f$ | 0.6$^{gh}$ | o.36$^{ef}$ | 0.59$^{efg}$ | 3.5$^{gh}$ | 32.8$^{ij}$ | 16.36$^{hi}$ | 85.0$^c$ |
| | T$_1$ | 0.80$^f$ | 0.64$^g$ | 0.88$^e$ | 0.63$^{efg}$ | 3.07$^{hi}$ | 31.63$^{ijk}$ | 16.1$^{ij}$ | 87. 25$^{ab}$ |
| | T$_2$ | 1.02$^f$ | 0.65$^g$ | 0.95$^e$ | 0.74$^e$ | 2.66$^{ij}$ | 32.24$^{ijk}$ | 15.8$^{ij}$ | 88.0$^a$ |
| | T$_3$ | 1.08$^f$ | 1.38$^{fg}$ | 1.57$^d$ | o.79$^e$ | 2.44$^j$ | 32.87$^i$ | 15.57$^{ij}$ | 88.07$^a$ |
| I$_2$ Moderate evaporation | T$_0$ | 1.88$^e$ | 2.0$^f$ | 1.67$^d$ | 1.28$^d$ | 5.9$^{bc}$ | 38.56$^h$ | 22.8$^c$ | 75.62$^g$ |
| | T$_1$ | 2.03$^e$ | 2.08$^f$ | 1.86$^d$ | 1.31$^d$ | 5.29$^{ed}$ | 43.6$^f$ | 21.9$^{bc}$ | 78.45$^f$ |
| | T$_2$ | 2.82$^{cd}$ | 3.15$^d$ | 2.69$^c$ | 2.10$^c$ | 4.34$^f$ | 45.64$^{ef}$ | 18.14$^{fg}$ | 80.05$^e$ |
| | T$_3$ | 3.38$^{bc}$ | 3.56$^c$ | 2.87$^c$ | 2.18$^c$ | 3.72$^g$ | 49.07$^d$ | 17.34$^{gh}$ | 81.98$^d$ |
| I$_3$ Severe evaporation | T$_0$ | 1.9$^e$ | 2.75$^e$ | 2.84$^c$ | 1.96$^c$ | 7.30$^a$ | 41.0$^g$ | 32.68$^a$ | 64.09$^i$ |
| | T$_1$ | 2.15$^e$ | 2.85$^e$ | 3.12$^{bc}$ | 1.90$^c$ | 6.34$^b$ | 43.52$^f$ | 27.44$^b$ | 65.32$^i$ |
| | T$_2$ | 3.63$^b$ | 4.07$^b$ | 3.85$^b$ | 2.95$^b$ | 5.57$^{cd}$ | 63.36$^b$ | 22.08$^{cd}$ | 70.29$^h$ |
| | T$_3$ | 4.19$^a$ | 4.39$^a$ | 4.44$^a$ | 3.44$^a$ | 5.72$^{cd}$ | 67.62$^a$ | 21.4$^{de}$ | 74.49$^g$ |
| F $_{test}$ | I × T | 0.612** | 0.49** | 0.317** | 0.34** | 0.509** | 99.01** | 14.49** | 14.22** |

Note: I$_0$ = 60mm evaporation, I$_1$ = 100mm evaporation, I$_2$ = 120mm evaporation, I$_3$ = 150mm evaporation, T$_0$ = No treatment, T$_1$ = NPK, T$_2$ = PGRs, T$_3$ = (T$_1$+T$_2$). The alphabetical superscript in a column present significant difference among the treatments to different rate of evaporation from highest to lowest value (a = highest value).

* = least significant

**significant

*** highly significant.

and mild irrigation (I$_1$) stress except lipid peroxidation and POX activity. The T$_1$ treatment exhibited non-significant difference on enzyme activities but the MDA content influenced significantly under all irrigation levels (I$_0$-I$_3$). The treatment T$_2$ (PGRs) and T$_3$ (NPK and PGRs) elevated the activities SOD, PPO, POX and SOD but reduced the MDA contents as compared to T$_0$ under severe water deficit (I$_4$) condition. (Table 7).

The interaction of irrigation intervals (I) and elicitors (T) significantly affect the membrane stability index (MSI) of canola (p<0.01). The water stress significantly reduced the membrane stability index (1–26%). NPK did not improve MSI while the combined application (T$_3$) significantly improve the MSI under moderate (I$_2$) and severe water stress (I$_3$) (Table 7).

## 4. Discussion

The findings of the current work highlighted the comparison of combined and individual application of PGRs and NPK to enhance canola growth, nutritional quality and yield under growing concerns of water scarcity. Particularly moderate to severe drought stress (I$_2$-I$_3$) imparts drastic effect on the canola growth by inducing injuries at all growth stages. It influenced the various morphological (reduced leaf growth, leaf area, plant height, number of nodes per plant), physiological traits (chlorophyll content, leaf water contents, leaf temperature and stomatal conductance) at the onset of water scarcity (Tables 3 and 4). Reduction of plant height was recorded in canola under different irrigation regimes compared to exogenously applied PGRs and NPK (Table 3). Growth retardation due to excessive evaporation, can be related to disruption of photosynthetic machinery and decline in carbon reserves for relocation to growing parts of plant [42,43]. It seems that the decrease in plant height also interferes

with leaf area. This reduction is particularly noticeable during post vegetative stage, flowering stage or abscission [43]. The grain yield reduced due to water stress as reported in pervious study [44]. Seed quality and 100 seed weight are commercial as well as economic traits, significantly compromised with ongoing scenario of water scarcity, and so is in the present experiment (Table 3). Seed filling is particularly influenced by drought stress by modulating of various metabolic activities occurring in the leaves, such as synthesis and translocation of photoassimilates, essential substrates for biosynthesis of seed storage reserves, mineral nutrients and many more functional constituents [45].

In the current experiment, drought led to reduction in chlorophyll content, and this loss could be due to some devastating effects on photosynthetic apparatus (Table 4). Stomatal conductance ($g_s$) was severely hampered when plants were exposed to severe rate of evaporation ($I_3$). The resistance in stomatal conductance ($g_s$) may be correlated to enhanced production of ABA under drought stress, which leads to stomata closure. ABA signaling mechanism tries to prevent the loss of tissue turgor by closing the stomata [46,47]. The optimal use of chemical fertilizers and PGR appreciably enhanced stomatal conductance under drought stress (Table 4). Yan et al. [48] also reported diffusional restrictions of $CO_2$ by stomata (52%), which directly caused a reduction of chlorophyll contents (31%) induced by drought. The fertilizer application especially urea increase the N supply at flowering and pod filling stage, delay leaf aging, enhanced chlorophyll contents and photo assimilates [49]. The total amount of chlorophyll contents increased by the availability of nitro compounds in the rhizosphere and consequently to the plants, ultimately produced more assimilates via photosynthesis which directly related to improve growth and yield [10,50]. Growth regulators and chemical fertilizers were significantly effective in mitigating the drastic effects of drought by maintaining the water efficiency of canola plants and augmenting the accretion of osmolytes. Accumulation of osmolytes may also favors the improvement of photosynthetic and gas exchange attributes [16]. The observed increase in yield of canola using NPK and PGRs under water limitation may be attributed to enhanced activities of CAT, SOD, POX and PPO [51]. Moreover, combined application of chemical fertilizer, PGRs and vermicompost particularly enhanced the accretion of secondary metabolites such as proline and sugar content and also chlorophyll synthesis [52].

Canola oil is the commercial commodity, while its content, profile and composition are affected by drought stress as reported in the current work (Table 5). Seed oil stems mostly from photosynthesis and green silique walls, later carbon is routed through different metabolic pathways into triacylglycerol occurring in the chloroplast, cytosol, and endoplasmic reticula [53]. The current experiment suggests 3–11% decrease in oil content and oil yield in the *B. napus* when the plants were exposed to irrigation stress, but Aslam et al. [54] reported a mere reduction of 3.2%.

The NPK treatment increases the flow of nutrient to the aerial part of plants and reduced the impact of water stress (Table 6). Due to water scarcity and loss of ionic balance, nutrients remained bond to the soil particles that are critical for the normal growth and development [6,55]. Same findings have been reported in the present work in the form of decreased biological yield (40%) with non-availability of nutrients (5–10%). Particularly, N, P and K contents increased with foliarly applied PGRs followed by chemical fertilizers in canola. This might be accredited to the role of K in biochemical pathways in plants. Potassium has a positive effect on metabolic processes of nucleic acids and proteins [56]. Phosphorus as a constituent of cell nuclei is essential for cell division and development of meristematic tissue of cotton. Further, P has a well-known impact in photosynthesis as well as synthesis of nucleic acids, proteins, lipids and other essential compounds [57]. The percentage of NPK uptake enhanced with the combined application of NPK and PGRs (Table 5). The bio fertilizers improve the soil textures and

bacterial colonization with the modification of physio-chemical properties of rhizosphere. On the other hand, the PGRs increased the photo assimilates translocation to sink (root tissue) and also improved the nutrient uptake and absorption power under water deficit and adverse environment condition. The PGRs can enhanced the activity of some vital N, P and K metabolizing enzymes in plant which enhanced NPK contents under different irrigation regimes [50,58].

All plants have been equipped with innate antioxidant enzyme mechanisms for the detoxification of reactive oxygen species. CAT decomposes $H_2O_2$ into water and molecular $O_2$. POX converts $H_2O_2$ by oxidizing co-substrates such as phenolic compounds and/or antioxidants and PPO in turn oxidizes phenols to chinone [59].

Membrane lipid peroxidation is a frequently used indictor to test the degree of plant sensitivity to oxidative damage caused by stress [60]. Our data revealed sensitivity of canola towards drought by elevation MDA content compared to $F_2$, $F_3$ treatments (Table 7). A lower level of lipid peroxidation presented high membrane stability (Table 7). It seems that the cell membrane integrity was maintained with chemical fertilizers and PGRs against the oxidative stress induced by water stress (Table 7). Mamnabi et al. [10] also suggested an amplification of antioxidant enzymes, total soluble sugars, photosynthetic attributes, leaf water content, membrane stability index and stomatal conductance but decreased the leaf temperature under different irrigation regimes.

The affirmative role of fertilizers and PGRs increase the antioxidant enzyme activities under water deficient condition in canola [61]. The PGRs treated plant showed more activities of antioxidant enzymes like POD, CAT and PPO as compared to untreated plants under moderate and severe water stress. The highest antioxidant activities were observed in $T_3$ treatments under severe stress as compared to $T_1$ and $T_2$ (Table 7). This supremacy was attained by the additive effect of fertilizer and PGRS on canola plants.

## 5. Conclusion

Canola is an emerging and unique oil seed crop among the other oil producing plants due to low erucic acid and glucosinolate contents. Here, we initially illustrate how NPK and PGRs, either individually or in combination, impact canola growth, photosynthetic and antioxidant activities and later seed yield and quality, and also attempt to explain its interaction to water scarcity for addressing these vital challenges. From the outcomes of current study, it appears rational to recommend chemical fertilizers (NPK) and PGRs (IAA and $GA_3$), that brought about better impact on canola seed yield, seed protein content, oil, oil composition with low glucosinolate and erucic acid contents, even under severe rate of evaporation (150mm). The harmful effects of stress were minimized considerably by the combined application of fertilizers and PGRs, thus improved growth attributes, chlorophyll content, MSI, stomatal conductance, antioxidants, osmoprotectants, grain yield and importantly, leading to a reduction in lipid peroxidation, particularly under moderate and severe rate of evaporation. These supremacies were attained by additive effects of NPK and PGPR, reducing the impact of drought. Such models can improve the probability of forecasting canola aptitude in challenging climates with an immediate response, but will also broadly help to select traits that can be further exploited through gene mining to produce sustainable and climate-resilient canola genotypes with considerable yield under high rate of water evaporation.

## Supporting information

**S1 Dataset.**
(XLSX)

## Author Contributions

**Conceptualization:** Muhammad Mahran Aslam.

**Data curation:** Muhammad Mahran Aslam, Fozia Farhat, Shafquat Yasmeen, Mahboob Ali Sial.

**Formal analysis:** Muhammad Mahran Aslam, Fozia Farhat, Mohammad Aquil Siddiqui, Muhammad Tahir Khan.

**Investigation:** Mohammad Aquil Siddiqui.

**Methodology:** Fozia Farhat, Mohammad Aquil Siddiqui, Shafquat Yasmeen.

**Supervision:** Mohammad Aquil Siddiqui.

**Writing – original draft:** Muhammad Mahran Aslam.

**Writing – review & editing:** Muhammad Mahran Aslam, Fozia Farhat, Mohammad Aquil Siddiqui, Muhammad Tahir Khan, Mahboob Ali Sial, Imtiaz Ahmad Khan.

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
