## [Decision Letter · Decision Letter 0]

21 Jul 2021

PONE-D-21-20309

Physiological and Biochemical Exploration of Canola for Overcoming Yield Limitation by Integrated Fertilizer management under different water regimes

PLOS ONE

Dear Dr. Siddique,

Thank you for submitting your manuscript to PLOS ONE. After careful consideration, we feel that it has merit but does not fully meet PLOS ONE’s publication criteria as it currently stands. Therefore, we invite you to submit a revised version of the manuscript that addresses the points raised during the review process. 

We look forward to receiving your revised manuscript.

Kind regards,

Khawaja Shafique Ahmad, Ph.D.

Academic Editor

PLOS ONE

Journal Requirements:

3. Please include your tables as part of your main manuscript and remove the individual files. Please note that supplementary tables (should remain/ be uploaded) as separate "supporting information" files

Reviewers' comments:

Reviewer's Responses to Questions

**Comments to the Author**

1. Is the manuscript technically sound, and do the data support the conclusions?

Reviewer #1: Yes

Reviewer #2: Yes

2. Has the statistical analysis been performed appropriately and rigorously? 

Reviewer #1: Yes

Reviewer #2: Yes

3. Have the authors made all data underlying the findings in their manuscript fully available?

Reviewer #1: Yes

Reviewer #2: Yes

4. Is the manuscript presented in an intelligible fashion and written in standard English?

Reviewer #1: Yes

Reviewer #2: No

5. Review Comments to the Author

Reviewer #1: This manuscript explores the physiological and biochemical response of canola against different water regimes. Moreover, the authors used three different treatments on canola genotypes subjected to four water regimes. The treatments also included chemical fertilizer as well plant growth regulators.

I found this MS quite interesting because of its research objectives. The physiological and biochemical analysis, authors did is also very much of interest. Therefore, I would like to recommend this article for publication, but after a minor revision. I noticed some issues in the article that should be corrected before acceptance.

Details are as following:

- Authors must provide the full form of any abbreviation used in the study at first usage. For instance, the full form of MDA was not provided at the first use. Kindly confirm the same for all other abbreviations used in the article.

- The authors must use line numbers while submitting an article to the journal. It is difficult for any reviewer to mention the specific desired changes without line numbers.

- I also see various spacing issues in the article. For instance, have a look at Page 3; Paragraph 2. Please fix all such issues throughout the MS.

- Check the citation of Table 7 in the article and correct it.

- I assume that there is a typo in the title of Table 1. Please fix it.

- The authors should also make sure that the text font is the same throughout. I found certain instances where the font was not the same.

- The acknowledgements section is not in fact presenting “Acknowledgements”. The section describes author contributions instead. Therefore, its title should be revised, and the Acknowledgements section should be written if there are any funding agencies/collaborators to be acknowledged concerning this research.

Regarding study design, novelty, and contribution to the field, I find this article suitable for publication and suggest a minor revision.

Thank you.

Reviewer #2: I would like to express appreciation to the authors for this study, I found the manuscript very interesting to read, but also came across few issues that will deserve some careful attention.

1. The author didn’t added line number , so it is very difficult to comment on some specific parts or lines where have some issues, generally I have added some comments in the PDF file, please carefully read the comments and similarly revise the whole MS.

2. The article need to be revised fully, precisely focusing on the results presentation, discussion and conclusion.

3. Abstract should be revised carefully and adopt the scientific style …..

4. The article should be revised logically to improve the results and discussion section particularly, conclusion is not good, should be revised.

5. I would recommend that all the table results should be used to draw figures to demonstrate the results significantly.

6. Reference style is not in one standard format, revise it and follow the journal style.

7. Overall, I recommend the article should be published after major revision of the papers results and English improvement.

6. PLOS authors have the option to publish the peer review history of their article (what does this mean?). If published, this will include your full peer review and any attached files.

Reviewer #1: No

Reviewer #2: No

---

## [Author Response · Author response to Decision Letter 0]

26 Sep 2021

File attached and response the all reviewer comments

---

## [Decision Letter · Decision Letter 1]

1 Nov 2021

PONE-D-21-20309R1Physiological and Biochemical Exploration of Canola for Overcoming Yield Limitation by Integrated Fertilizer management under different water regimesPLOS ONE

Dear Dr. Aquil,

Thank you for submitting your manuscript to PLOS ONE. After careful consideration, we feel that it has merit but does not fully meet PLOS ONE’s publication criteria as it currently stands. Therefore, we invite you to submit a revised version of the manuscript that addresses the points raised during the review process. I would expect your fully revised manuscript with point by point response to the reviewers comments within due date. 

ACADEMIC EDITOR: 

 I would suggest to read carefully the entire MS and revise the English grammar, abbreviation and formatting. Moreover, strictly follow the reviewers suggestions and address all the comments.

We look forward to receiving your revised manuscript.

Kind regards,

Khawaja Shafique Ahmad, Ph.D.

Academic Editor

PLOS ONE

Reviewers' comments:

Reviewer's Responses to Questions

**Comments to the Author**

1. If the authors have adequately addressed your comments raised in a previous round of review and you feel that this manuscript is now acceptable for publication, you may indicate that here to bypass the “Comments to the Author” section, enter your conflict of interest statement in the “Confidential to Editor” section, and submit your "Accept" recommendation.

Reviewer #2: All comments have been addressed

Reviewer #3: (No Response)

2. Is the manuscript technically sound, and do the data support the conclusions?

Reviewer #2: Yes

Reviewer #3: Partly

3. Has the statistical analysis been performed appropriately and rigorously? 

Reviewer #2: Yes

Reviewer #3: Yes

4. Have the authors made all data underlying the findings in their manuscript fully available?

Reviewer #2: Yes

Reviewer #3: Yes

5. Is the manuscript presented in an intelligible fashion and written in standard English?

Reviewer #2: Yes

Reviewer #3: No

6. Review Comments to the Author

Reviewer #2: Review of ““Physiological and Biochemical Exploration of Canola for Overcoming Yield Limitation by Integrated Fertilizer management under different water regimes” for PolsOne.

This manuscript describes a study entitled “Physiological and Biochemical Exploration of Canola for Overcoming Yield Limitation by Integrated Fertilizer management under different water regimes”.

I would like to express my appreciation to the authors for revising the MS, I found that manuscript has been improved and much better than earlier version, but also came across few issues that still deserve some careful attention. I have highlighted few places and would suggest to read carefully the entire MS and revise the English grammar, abbreviation and formatting. Overall, I recommend the article should be published after revision of these minor issues to improve the quality of their work.

Reviewer #3: I can see that the authors have revised the manuscript in the light of reviewer comments. However, more information is required and it still needs an extensive revision before acceptance for publication in the journal. My suggestions are given below:

The title does not truly reflect the study and may be revised as: "Effects of exogenous NPK and plant growth regulators application on physiological and biochemical processes of canola under drought stress".

Abstract: "Environmental stressors" or "Environmental stresses"? Replace "modify" with "alter".

Write "phyto-hormones" as "phytohormones"

Chemical fertilizers are not a natural source. Also, the phytohormones are exogenously applied in this study so is it appropriate to mention them as a natural source here?

"key interest" or "major aim"?

What is seed stage? Please elaborate and correct.

Mention the sources and rate of application of NPK fertilizers in the abstract.

Which plant growth regulators? At which crop growth stage were they applied? Briefly mention here in the abstract.

The results are not well described in abstract and are too general. Try to be more specific in terms of treatments and also give percent increase or decrease in each observation by the application of treatments.

In key words, delete "elicitors", "mitigation strategies". Correct "water shortage" as "water stress". Also, include scientific name of canola and the names of plant growth regulators used in this study.

Introduction:

"withhold" or "holds"?

Correct "With constant efforts of scientists" as "Owing to continuous efforts of scientists..."

The statistics related to area and yield of canola are related to which country? Also give the year of these statistics.

"inefficient" or "poor"?

Correct "management of macro and micronutrient" as "management of macro- and micronutrients". Also is this the only reason for low yield of canola in Pakistan?

Rewrite "environmental stress alone and in combinations of multiple stresses, such as heat and drought" as " different environmental stresses, particularly heat and drought".

Which physiological and biochemical mechanisms are affected by loss in turgor? Briefly explain here. Also check the grammar of this sentence.

Delete "caused the" before "The water deficit condition". Instead write "the" before "intercellular".

"The water deficit condition reduced the chlorophyll destruction, photochemical system disorder and stomatal closure"? Recheck this sentence.

"The plant showed physiological response in the form of reactive oxygen species (ROS) production under drought stress". Is it a physiological or biochemical response? Also, correct "The plants showed......" as "The plants respond to drought........"

Correct as "Nitrogen plays a pivotal role....". Use "N" for "nitrogen" at first mention and follow this abbreviation throughout the manuscript. The same holds for P and K.

"decrease" instead of "sway".

Always use "-" after micro- when mentioning both together as micro- and macro nutrients....

Correct "This lead to improve nutrient uptake and reduce impact of drought in crops" as "It improves nutrient uptake and reduces the damaging effects of drought in crops".

Rewrite this sentence with proper grammar "but current world situations required some immediate response methods for the

the growing demands of food and feed"

The hypothesis and aims of the study should be given in the introduction.

Materials and Methods

Correct as "Two year field experiments were....."

Rewrite this sentence "in reaction to fertilization under excessive rate of water evaporation". "response" instead of "reaction" Also, rate of evaporation or drought stress was one of the main factors of this study?

The treatment details should be either given in a table or as a paragraph.

GA and IAA should be defined at first mention.

The M&M lacks information such as:

Are these the recommended NPK rates for canola? What was the reason for the selection of these NPK application rates? Were all the fertilizers including N applied at the time of sowing?

Which canola cultivar was used in this study. The data about the soil properties as well as the climatological data should be the part of M&M instead of results.

How did the authors apply this small amount of IAA and GA to the plants in the field. At which canola growth stage were the foliar treatments carried out? What was the control for foliar spray treatments? Did they use any surfactant for foliar treatments?

Give the instrumental settings for the measurement of stomatal conductance.

How did the authors manage the different irrigation regimes?

Line 177: Correct 25oC. Check throughput the manuscript.

Line 176: What was the pH of the phosphate buffer used for extraction?

Line 180: Correct as "Units/g FW". Needs to be consistent about the use Units/g OR Units g-1 throughout the manuscript.

Line 181: Correct as "interval of 30..."

Line 185: "affected" instead of "effected". Also correct as "nitroblue tetrazolium"

Line 189: Avoid starting as sentence with a number i.e. 0.5 here

The results are poorly written and should give percent increase or decrease in each attribute by the application of treatments. Also, there are many typing and grammatical mistakes which should be carefully corrected, for example, write p < 0.01 as p<0.01. Check spelling and grammar throughout the manuscript.

The literature given in the discussion is outdated, for instance, the authors have reported studies of 1999, 2001, 2007 etc. More recent literature should be included, preferably not older than last five years. Also, the discussion should include how the observed increase in yield and quality attributes is linked to various physiological and biochemical responses under drought stress. What was the reason behind the observed increments by the application of GA and IAA in water stressed canola plants?

In Table 3, correct the units for grain yield. I would suggest to write in full. Also correct "Kg/ha" as "kg/ha".

The manuscript needs extensive revision before it is considered for publication.

7. PLOS authors have the option to publish the peer review history of their article (what does this mean?). If published, this will include your full peer review and any attached files.

Reviewer #2: No

Reviewer #3: No

---

## [Editor Report · Decision Letter 2]

22 Nov 2021

Exploration of Physiological and Biochemical Processes of Canola with Exogenously Applied Fertilizers and Plant Growth Regulators under Drought Stress

PONE-D-21-20309R2

Dear Dr. Aquil,

We’re pleased to inform you that your manuscript has been judged scientifically suitable for publication and will be formally accepted for publication once it meets all outstanding technical requirements.

Kind regards,

Khawaja Shafique Ahmad, Ph.D.

Academic Editor

PLOS ONE
---

## [Editor Report · Acceptance letter]

26 Nov 2021

PONE-D-21-20309R2 

Exploration of Physiological and Biochemical Processes of Canola with Exogenously Applied Fertilizers and Plant Growth Regulators under Drought Stress 

Dear Dr. Siddiqui:

I'm pleased to inform you that your manuscript has been deemed suitable for publication in PLOS ONE. Congratulations! Your manuscript is now with our production department. 

Kind regards, 

on behalf of

Dr. Khawaja Shafique Ahmad 

Academic Editor

PLOS ONE